# RAC1B Induces SMAD7 via USP26 to Suppress TGFβ1-Dependent Cell Migration in Mesenchymal-Subtype Carcinoma Cells

**DOI:** 10.3390/cancers12061545

**Published:** 2020-06-11

**Authors:** Hendrik Ungefroren, Anuradha Kumarasinghe, Melina Musfeldt, Christian Fiedler, Hendrik Lehnert, Jens-Uwe Marquardt

**Affiliations:** 1First Department of Medicine, University Hospital Schleswig-Holstein, Campus Lübeck, D-23538 Lübeck, Germany; anu.95@hotmail.de (A.K.); melinaa.musfeldt@gmail.com (M.M.); c.fiedler93@gmx.de (C.F.); Jens.Marquardt@uksh.de (J.-U.M.); 2Clinic for General Surgery, Visceral, Thoracic, Transplantation and Pediatric Surgery, University Hospital Schleswig-Holstein, Campus Kiel, D-24105 Kiel, Germany; 3University of Salzburg, A-5020 Salzburg, Austria; hendrik.lehnert@sbg.ac.at

**Keywords:** cell migration, CRISPR/Cas9, pancreatic ductal adenocarcinoma, RAC1B, SMAD7, TGFβ, triple-negative breast cancer, USP26

## Abstract

The small GTPase RAC1B has been shown to act as a powerful inhibitor of the transforming growth factor (TGF)β type I receptor ALK5 and TGFβ1/ALK5-induced epithelial–mesenchymal transition and cell motility. However, the precise mechanism has remained elusive. RNAi-mediated knockdown of RAC1B in the pancreatic ductal adenocarcinoma (PDAC)-derived cell line Panc1 failed to alter transcriptional activity from a transfected ALK5 promoter–reporter construct. In contrast, pharmacological inhibition of the proteasome decreased the abundance of ALK5 protein in cell lines of the mesenchymal subtype (Panc1, IMIM-PC-1, and breast cancer MDA-MB-231), but not in a PDAC cell line of the epithelial subtype (Colo357). Here, we focused on the inhibitory Smad protein, SMAD7, as a potential candidate for RAC1B-mediated inhibition of cell migration. In Panc1 cells devoid of RAC1B, SMAD7 protein was dramatically reduced and these cells were refractory to TGFβ1-induced upregulation of SMAD7 protein but not mRNA expression. Intriguingly, RNAi-mediated knockdown or ectopic overexpression of SMAD7 in Panc1 cells up- or downregulated, respectively, ALK5 protein expression and mimicked the suppressive effect of RAC1B on TGFβ/SMAD3-dependent transcriptional activity, target gene expression and cell migration. Transfection of SMAD7 was further able to partially rescue cells from the RAC1B knockdown-mediated increase in migratory properties. Conversely, knockdown of *SMAD7* was able to partially rescue Panc1 and MDA-MB-231 cells from the antimigratory effect of ectopically expressed RAC1B. Finally, we demonstrate that RAC1B upregulation of SMAD7 protein requires intermittent transcriptional induction of the deubiquitinating enzyme USP26. Our data suggest that RAC1B induces SMAD7 by promoting its deubiquitination and establishes this Smad as one of RAC1B’s downstream effectors in negative regulation of ALK5 and TGFβ1-induced cell migration in mesenchymal-type carcinoma cells.

## 1. Introduction

Pancreatic ductal adenocarcinoma (PDAC) and triple-negative breast cancer (TNBC) are among the most aggressive and early metastasizing tumors [1,2]. The predominant genomic alterations in PDAC and TNBC affect the *KRAS* oncogene and the tumor suppressor gene *DPC4*. PDAC and TNBC represent highly heterogeneous diseases characterized by diverse molecular and morphological features and poor treatment response. The major reason for TNBC poor prognosis is early therapeutic escape from conventional treatments, leading to aggressive metastatic relapse. Recent genomic and transcriptomic profiling of surgically resected human tumors revealed different evolutionary routes leading to PDAC and highlighted the existence of several, in part overlapping, subtypes [3,4,5], which reflect both tumor cell-intrinsic and microenvironment-specific features. For PDAC, these subtypes have been variably designated as “classical” and “quasi-mesenchymal” [3], or “basal-like”, “squamous”, “pancreatic progenitor”, “immunogenic” and “aberrantly differentiated endocrine exocrine” (ADEX) [4], and for TNBC basal-like-1/2, immune-modulatory, luminal androgen receptor, mesenchymal, mesenchymal/stem-like subtype and claudin-low [2,6]. A comparison of PDAC subtypes based on histomorphological characterization with molecular subtyping identified a partial overlap with previously described transcription-based classifications. Differentiated tumors (G1/G2) were enriched in gene expression signatures from the previously reported classical subtype, while undifferentiated tumors (G3) were characterized by enrichment in basal-like, squamous and quasi-mesenchymal signatures.

In both PDAC and TNBC, metastases occur after an epithelial–mesenchymal transition (EMT) of epithelial cells, allowing them to dissociate from the primary tumor site and to colonize distant organs [7]. Moreover, it has been shown that transcriptional subtypes are associated with different modes of EMT, whereby poorly differentiated transcriptional subtypes correlate with “complete EMT” (c-EMT, represented by the PDAC-derived cell lines Panc1 and IMIM-PC-1, and the TNBC-derived cell line MDA-MB-231), and well-differentiated transcriptional subtypes correlate with “partial EMT” (p-EMT, represented by the PDAC-derived cell line Colo357). Rather than by transcriptional repression as in c-EMT tumors, those of the p-EMT subtype lose their epithelial phenotype through an alternative program involving protein internalization and the action of members of the Rab subfamily of GTPases [8]. The least differentiated quasi-mesenchymal (PDAC) or mesenchymal and mesenchymal/stem-like (TNBC) tumors generally indicate the subgroups, which are enriched for genes involved in EMT, invasion and metastasis and are characterized by worse prognosis and resistance to standard chemotherapy [5,9]. In order to overcome their malignant behavior and eventually chemoresistance, a more thorough characterization of the specific features that distinguish poorly differentiated mesenchymal tumors from their less malignant and chemoresistant well-differentiated counterparts is mandatory. Aberrantly activated signals found in different subgroups of PDAC and TNBC include those of the Wnt/β-catenin, Notch, Sonic Hedgehog, and TGFβ signaling pathways. TGFβ signaling plays essential roles in multiple development stages of PDAC and TNBC tumors as well as in chemo- and radioresistance through its ability to regulate stemness, EMT, apoptosis [7,10], and the DNA damage response [11].

Binding of TGFβ to its cognate receptors, the TGF-β type II receptor and the TGF-β type I receptor, ALK5, results in activation of canonical Smad signaling involving C-terminal phosphorylation of receptor-regulated Smads (R-Smads) SMAD2/3, complex formation with SMAD4 and transcriptional regulation of TGFβ target genes. Activated ALK5 is known to rapidly induce in a TGFβ/SMAD2/3-dependent manner the expression of the inhibitory Smads, SMAD6 or SMAD7. Whereas SMAD6 preferentially inhibits SMAD signaling initiated by the bone morphogenetic protein (BMP) type I receptors, SMAD7 interferes with both TGFβ and BMP-induced Smad signaling [12] through multiple mechanisms in the cytoplasm and in the nucleus. For instance, SMAD7 can form stable complexes with activated type I receptors, thereby blocking the phosphorylation of R-Smads, or recruit ubiquitin E3 ligases, such as Smurf1/2, resulting in the ubiquitination and degradation of the activated type I receptors [13,14]. As is the case for ALK5, SMAD7 itself is also targeted for ubiquitination and proteasome-mediated degradation by E3 ligases [15]. This process is antagonized by deubiquitinating enzymes (DUBs) and several ubiquitin-specific proteases (USPs) such as USP4 [16], 11 [17], 15 [18], 26 [15], UCH37 [19] and UCHL1 [20] have been implicated in TGFβ signaling [21].

In addition, the canonical Smad pathway TGFβ activates, or cross-talks with, several non-Smad signaling proteins including the small Rho-like GTPase RAC1 and its less prominent splice isoform, RAC1B [22,23]. RAC1B differs from RAC1 by an in-frame insertion of an additional exon, exon 3b, which encodes for 19 amino acids. This insertion behind the switch II region of RAC1 results in an impaired hydrolysis of GTP and an accelerated exchange of GDP to GTP ([24] and references therein). Recent data suggest that the inclusion of exon 3b alters specific biochemical and signaling properties of RAC1B that in specific cellular contexts or cell types can result in functional antagonism to RAC1 [24]. For instance, while RAC1 is known to promote TGFβ1-induced EMT and cell migration/invasion, RAC1B inhibits both responses in benign and malignant pancreatic [22,23] and breast [25,26] epithelial cells. Mechanistic studies revealed that RAC1B was required to maintain an epithelial gene expression profile while suppressing a mesenchymal program [27]. Moreover, RAC1B expression correlated with the differentiation status/tumor subtype being high in well-differentiated cell lines with epithelial/classical phenotype and low in poorly differentiated lines with (quasi-)mesenchymal appearance [27]. This raises the exciting possibility that RAC1B is causatively involved in the subtype conversion from epithelial to mesenchymal. Although the mechanisms governing this transition in vivo remain elusive it is likely driven by one or more of the above mentioned signaling pathways, in particular TGFβ. In this context, it is noteworthy that RAC1B potently inhibits the expression level of ALK5 [23,28,29], which is at the core of the cells’ sensitivity to this growth factor. However, the mechanistic basis of this crucial regulatory event is not known but may involve transcriptional silencing of *TGFRBI* or posttranscriptional regulation by inhibitors of the TGFβ pathway. Prompted by a prolonged activation of SMAD3 and p38 MAPK and enhanced chemokinetic activity in RAC1B-deficient cells following TGFβ1 stimulation [23,29], we addressed the question of whether altered regulation of SMAD7 is involved in these effects. We primarily employed the PDAC-derived cell line Panc1 and the TNBC-derived line MDA-MB-231—both of which are poorly differentiated mesenchymal-like cells though have retained a functional TGFβ/Smad pathway [21,22].

## 2. Results

### 2.1. Negative Regulation of ALK5 by RAC1B May Involve Changes in ALK5 Protein Stability Rather than Transcriptional Activity of TGFBR1

We have recently shown that cellular depletion of RAC1B led to an increase in both ALK5 protein abundance and TGFβ1-dependent responses [28], and we sought to elucidate the molecular basis underlying this crucial function of RAC1B. Since we also observed earlier a moderate induction of ALK5 mRNA following RAC1B knockdown [23], we initially pursued the idea that RAC1B mediates silencing of *TGFBRI* at the transcriptional level by blocking de novo transcription from its promoter. To investigate this possibility, we performed reporter gene assays with Panc1 cells and an ALK5 promoter–reporter fusion gene encompassing 392 bp upstream of the transcription start site (TβRI-392/+21-pGL4) [30]. As shown in Figure 1, the siRNA-mediated knockdown of RAC1B had no significant effect on the activity of this reporter relative to control transfectants (Figure 1A, upper graph). In contrast, transfecting Panc1 cells under the same conditions with the TGFβ/SMAD3-responsive reporter plasmid, p(CAGA)_12_-luc, and the same siRNA to RAC1B, but not a bi-specific siRNA targeting both RAC1B *and* RAC1, resulted in a large increase in luciferase activity (Figure 1A, lower graph).

ALK5 expression can also be regulated at the protein level by ubiquitination and deubiquitination [14]. The treatment of other mesenchymal cell lines (IMIM-PC-1, Panc1 and MDA-MB-231), but not an epithelial/classical cell line (Colo357), with the proteasome inhibitor MG132 resulted in concentration-dependent decrease in ALK5 protein levels (Figure 1B). In contrast, the ALK5 mRNA levels remained unaffected in these cells except for MDA-MB-231. However, here, the decline was less pronounced than that of the protein (Appendix A). The observation that ALK5 protein abundance in IMIM-PC-1, Panc1 and MDA-MB-231 cells was decreased rather than increased in response to proteasome inhibition suggests that ALK5 itself is unlikely to be the target of proteasomal degradation. Rather, RAC1B may regulate ALK5 by inducing an as yet unknown factor that is stabilized by proteasome inhibition to subsequently promote ALK5 degradation. Likely candidates for such a protein are the inhibitory Smads, SMAD6 and SMAD7. Both genes are TGFβ target genes that terminate R-Smad activation by the ALK5 kinase and eventually promote ALK5 degradation, and are themselves regulated by ubiquitination [14,15]. This provided a hint for possible involvement of SMAD7 in RAC1B regulation of ALK5. Since SMAD6 preferentially inhibits Smad signaling initiated by ALK3 and ALK6, we focused here on SMAD7. So far, we conclude that inhibition of *TGFBR1* promoter activity and de novo transcription is unlikely to be responsible for the suppressive effect of RAC1B on ALK5 expression. Rather, RAC1B’s effect on ALK5 may be indirect and involve intermittent induction or activation of a protein(s) that promotes ALK5 proteasomal degradation.

### 2.2. MG132 Increases SMAD7 and Decreases ALK5 Abundance in Panc1-RAC1B-KO Cells

Prompted by the sensitivity of ALK5 protein abundance to MG132 treatment we next analyzed whether SMAD7 protein levels in pancreatic epithelial cells are affected by proteasome inhibition. Specifically, we hypothesized that treatment of cells with MG132 should inhibit ubiquitin-mediated degradation of SMAD7 and, as a consequence, increase its steady-state levels. To verify this assumption, we employed Panc1 cells with high endogenous ALK5 expression as a result of deleting RAC1B-specific exon 3b of *RAC1* by CRISPR/Cas technology (Panc1-RAC1B-KO cells) [28]. The treatment of these cells with MG132 followed by evaluation of SMAD7 expression by immunoblotting showed that these cells display clearly elevated levels of SMAD7 protein (Figure 2A). Strikingly, the increase in SMAD7 expression upon MG132 treatment was associated with a dramatic decrease in ALK5 expression in Panc1-RAC1B-KO cells (Figure 2B). We conclude from these data that the MG132-mediated increase in the steady-state levels of SMAD7 was the result of its reduced proteasomal degradation and may be causally involved in the decrease in ALK5 abundance and eventually TGFβ1-induced cell migration.

### 2.3. The Depletion of RAC1B Reduces Basal and TGFβ-Induced SMAD7 Protein but not mRNA Expression

Given the reciprocal sensitivity of SMAD7 and ALK5 to proteasomal inhibition on the one hand and the elevated ALK5 expression in Panc1-RAC1B-KO cells [28] on the other hand, we pursued the idea that RAC1B maintains SMAD7 protein expression by either stimulating coupled transcription/translation or by increasing stability of the protein. According to this assumption, SMAD7 expression should be reduced in Panc1-RAC1B-KO cells compared to vector controls. To this end, basal levels of SMAD7 protein in Panc1-RAC1B-KO cells were only 27% of levels detected in controls (Figure 3A). Moreover, a 2 h treatment with TGFβ1 resulted in an approx. 2-fold increase in SMAD7 protein abundance in control cells, while Panc1-RAC1B-KO cells failed to upregulate SMAD7 protein in response to TGFβ1 (Figure 3B). Transient knockdown of RAC1B in MDA-MB-231 cells also resulted in downregulation of SMAD7 protein.

When we measured SMAD7 mRNA abundance in Panc1-RAC1B-KD and -KO cells treated for 1 h with TGFβ1, we noted that it was induced up to 5-fold at the 1 h time point. However, in contrast to protein expression, no differences were observed in RAC1B-KD vs. control cells, while in the RAC1B-KO cells the SMAD7 mRNA levels were even higher than in controls (Figure 3C), likely reflecting stronger activation of the pathway due to upregulation of ALK5. Since SMAD7 mRNA and protein expression in response to TGFβ1 stimulation did not correlate with each other and RAC1B-KO cells were unable to upregulate SMAD7 protein in response to TGFβ1, we assumed that RAC1B regulation of SMAD7 expression occurs primarily at the protein level.

### 2.4. SMAD7 Mimics the Inhibitory Effect of RAC1B on ALK5 Protein Expression, TGFβ1/Smad-Induced Transcriptional Activity and Target Gene Expression

We have previously shown that knockdown or knockout of RAC1B resulted in enhanced abundance of ALK5 protein [28] and an increase in TGFβ signaling [23,28]. Thus, we next explored if SMAD7 siRNA-mediated knockdown of SMAD7 is able to mimic these effects. Intriguingly, transient transfection of a SMAD7 siRNA but not a scrambled control siRNA increased ALK5 abundance in Panc1-LV cells (Figure 4A), while conversely, ectopic expression of SMAD7 but not empty vector in ALK5^high^ Panc1-RAC1B-KO cells strongly reduced the abundance of ALK5 protein (Figure 4B).

Given the ability of ectopic SMAD7 to mimic the negative effect of RAC1B on ALK5 abundance, we reasoned that SMAD7 should also be able to reproduce the effects of a knockdown or ectopic overexpression of RAC1B on TGFβ/SMAD3-dependent transcriptional activity (see Figure 1A and [22]) and expression of the TGFβ target genes *MMP2* and *SNAI2* [23]. To this end, knockdown of *SMAD7* in Panc1 cells using a mixture of two different pre-evaluated siRNAs enhanced SMAD3-dependent transcriptional activity (Figure 4C). Likewise, reducing the levels of SMAD7 enhanced TGFβ1-stimulated expression of *SNAI2* and *MMP2* mRNA in both Panc1 (Figure 4D) and MDA-MB-231 cells (Appendix A), but failed to do so in Colo357 cells (Appendix A). Conversely, Panc1 cells ectopically expressing SMAD7 presented with a decrease in transcriptional activity of the p(CAGA)_12_-luc reporter (Figure 4E), and *SNAI2* and *MMP2* expression (Figure 4F). This clearly indicates that SMAD7 can mimic the effect of RAC1B on transcriptional induction of individual genes involved in EMT and cell invasion. In good match with our results, siRNA-mediated knockdown of SMAD7 increased TGFβ-induced activation of SMAD3 in MDA-MB-231 cells [31]. In sum, these data show that SMAD7 mimics the inhibitory effect of RAC1B on ALK5 protein expression and provide further evidence for the assumption that RAC1B exerts its inhibitory effect on TGFβ/Smad-mediated gene expression at least in part through upregulation of SMAD7.

### 2.5. SMAD7 Knockdown Mimics the Stimulatory Effect of RAC1B Knockdown on TGFβ1-Induced Chemokinesis

RAC1B is a potent inhibitor of basal and TGFβ1-induced cell migration and hence its knockdown amplified the cells’ migratory potential [22,23]. To verify whether SMAD7 can duplicate this effect, we depleted cells of SMAD7 protein with the same combination of siRNAs used above. To this end, knockdown of SMAD7 enhanced basal and TGFβ1-induced cell migration (Figure 5A, left-hand graph) as did knockdown of RAC1B performed as control (Figure 5A, right-hand graph). Similar data, albeit displaying less dramatic effects, were obtained with MDA-MB-231 cells (Appendix A), in contrast to Colo357 cells in which the SMAD7 knockdown lacked an effect (Appendix A). Interestingly, codepletion of SMAD7 and RAC1B in Panc1 cells modestly enhanced the promigratory effect over that of the SMAD7 knockdown alone (Figure 5B).

We have previously characterized Panc1 cells with stable expression of a HA-tagged version of RAC1B (Panc1-HA-RAC1B) and have shown that these cells exhibit a decrease in TGFβ1-dependent cell migration [22]. To verify whether SMAD7 protein expression was responsible for the reduced migratory potential of these cells, we knocked down SMAD7 by RNAi in Panc1-HA-RAC1B cells and empty vector (pCGN) control cells and compared the cells’ migratory activity with that of irrelevant control siRNA-transfected cells. The SMAD7 knockdown potently relieved suppression of migratory activity in these cells (Figure 5C). Together, these data show that depleting cells of SMAD7 can mimic the promigratory effect of RAC1B knockdown on TGFβ1-dependent random cell migration. This adds further proof to our hypothesis that SMAD7 has a functional effector role in RAC1B-mediated suppression of cell motility.

### 2.6. Ectopic Overexpression of SMAD7 Mimics the Antimigratory Effect of RAC1B in Mesenchymal-Type but Not Epithelial-Type Carcinoma Cells

We have previously shown that Panc1-RAC1B-KO cells have strongly increased ALK5 expression which was associated with an enhanced migratory response to TGFβ1 stimulation [28]. Given the increase in SMAD7 and decrease in ALK5 abundance following MG132 exposure, we reasoned that the MG132 treatment should also impair the cells’ chemokinetic response. To this end, Panc1-RAC1B-KO and MDA-MB-231 cells subjected to real-time cell migration assay in the presence of TGFβ1 and MG132 had dramatically reduced chemokinetic activity compared to TGFβ1 + vehicle-treated cells (Appendix A).

The stimulatory effect of RAC1B on SMAD7 protein levels and the inhibitory effect of SMAD7 on ALK5 abundance suggested the possibility that the suppressive effect of RAC1B on TGFβ/ALK5-induced signaling and cell migration relies on its ability to maintain (high) SMAD7 expression. Hence, we postulated that ectopic expression of SMAD7 should mimic the antimigratory effect of RAC1B. To this end, transfection of MDA-MB-231 cells with SMAD7 (Figure 6A, left-hand graph) or HA-RAC1B (Figure 6A, right-hand graph), and Panc1 cells with SMAD7 (Figure 6B), but not empty vector, dramatically decreased TGFβ1-dependent chemokinetic activity in both cell types. In contrast, transfection of the well-differentiated cell line Colo357 with SMAD7 failed to repress TGFβ1-induced migration (Figure 6C).

### 2.7. Ectopic Overexpression of SMAD7 Rescues Cells from the RAC1B Knockdown-Induced Increase in Migration

In Figure 6, we have evaluated the effects of ectopic SMAD7 expression on otherwise non-genetically engineered cells. However, we postulated that ectopic expression of SMAD7 should also be able to reverse in Panc1 cells both the RAC1B knockdown and knockout-mediated increase in TGFβ1-stimulated chemokinetic activity. As depicted in Figure 7, ectopic SMAD7 strongly and moderately inhibited the migratory of Panc1-RAC1B-KD (Figure 7A) and KO (Figure 7B) cells, respectively. Hence, we conclude that RAC1B induces SMAD7 to decrease the abundance of ALK5 and, as a consequence, the sensitivity of cells to TGFβ1-induced cell migration.

### 2.8. USP26 Mediates the Promoting Effect of RAC1B on SMAD7 Expression

The failure of TGFβ1 to upregulate SMAD7 protein but not mRNA abundance in RAC1B-depleted cells indicated that RAC1B regulates SMAD7 at the protein level. A possible scenario is the induction of a DUB that deubiquitinates SMAD7 to prevent its proteasome-mediated degradation. To test whether any of the USPs implicated before in the regulation of TGFβ signaling (USP4, 11, 15, and 26) are regulated by RAC1B, we profiled their expression in Panc1-RAC1B-KD and control cells by qPCR. Notably, mRNA levels of USP26, and to a lesser extent USP4, were lower in RAC1B-KD cells, while those of USP11 and 15 were not altered (Figure 8A). The USP26 mRNA was also strongly decreased in Panc1-RAC1B-KO cells (Figure 8B). Prompted by these findings and a recent study showing that USP26 directly targets SMAD7 [15], we focused our attention on this DUB in subsequent experiments. To reveal whether USP26 is induced by TGFβ in PDAC-derived cells, we analyzed its mRNA expression in Panc1 and Colo357 cells after short-term treatment with TGFβ1. As previously shown for MDA-MB-231 and other cell types [15], TGFβ1 rapidly enhanced the mRNA levels of *USP26* in Panc1 but not in Colo357 cells (Figure 8C). Next, we addressed the question of whether the response of *USP26* to TGFβ1 is also seen at the protein level and, if so, whether RAC1B impinges on this. To this end, treatment of Panc1-RAC1B-KO and LV control cells with TGFβ1 for 2 h resulted in upregulation of USP26 protein levels only in LV cells as shown by immunoblotting (Figure 8D).

Above, we have shown that SMAD7 mimics the effects of RAC1B on ALK5 abundance, TGFβ1/Smad-dependent reporter gene activity, gene expression, and cell migration. If USP26 functionally links RAC1B and SMAD7 then USP26, too, should be able to reproduce these effects of RAC1B. To find out, we knocked down USP26 by RNAi in Panc1 cells and evaluated the consequences on SMAD7 protein expression. Intriguingly, USP26 depletion severely impaired upregulation of SMAD7 by TGFβ1 (Figure 8E), and enhanced TGFβ1-induced transcriptional activity (Figure 8F) and gene expression of *MMP2* and *SNAI2* (Figure 8G). Finally, knockdown of *USP26*, or *SMAD7* as control, both strongly enhanced basal and TGFβ1-dependent migratory activity in Panc1 cells (Figure 8H). These data show that in Panc1 cells, *USP26* is transcriptionally induced by TGFβ1 in a RAC1B-dependent manner and mediates the effects of RAC1B on SMAD7 expression, and on TGFβ1/ALK5-induced transcriptional responses.

## 3. Discussion

In previous reports, we have demonstrated that RAC1B potently inhibited TGFβ1-dependent activation of SMAD2/3 [22] and p38 and ERK MAPK [23,27], growth inhibition [22,28], and cell migration [22,23,28,29]. In follow-up studies, we were able to show that RAC1B suppressed the protein (and mRNA) expression of the TGFβ type I receptor, ALK5 [28], via intermittent downregulation of the G protein-coupled receptor proteinase-activated receptor 2 [32]. However, the mechanistic basis of the changes in ALK5 abundance and corresponding changes in its kinase activity has remained elusive.

RAC1B has been reported to be localized in the nucleus [33], a finding that would principally be compatible with a role as a transcription factor and, hence, direct (inhibitory) transcriptional effects of RAC1B on the *TGFBR1* promoter. However, reporter gene assays showed that knockdown of RAC1B failed to alter transcriptional activity of a cotransfected reporter construct encompassing 392 base pairs upstream of the transcriptional start site in the *TGFBR1* promoter. The same intervention, however, was able to increase activity of the TGFβ/SMAD3-responsive reporter p(CAGA)_12_-luc (see Figure 1A), suggesting that the ALK5 promoter is unresponsive to repression by RAC1B. This is in line with earlier findings revealing that the increase in ALK5 mRNA following RAC1B knockdown was secondary to the rise in ALK5 protein abundance and may thus have been caused by ALK5 autostimulation [28]. The lack of transcriptional repression of *TGFBR1* by RAC1B did not, however, exclude other modes of negative regulation, as TGFβ signaling is known for its extensive negative (feedback) regulation. For instance, depending on cell type and cellular context, the ALK5 and SMAD2, 3, 4 and 7 proteins can all be regulated via ubiquitin-mediated proteasomal degradation [14]. Initial treatment of cells with the proteasome inhibitor MG132 indeed showed that the levels of ALK5 decreased in a dose-dependent manner in several poorly differentiated mesenchymal-type lines but not in well-differentiated epithelial-like Colo357 cells. However, ALK5 itself does not appear to be the (direct) target of this inhibitor, as this should result in an increase in protein abundance. Rather, we pursued the idea that MG132 prevents the degradation of a protein(s) that negatively regulates the stability of ALK5, such as SMAD7. The present study was therefore initiated to elucidate the role of SMAD7 in mediating the suppressive function of RAC1B on ALK5 and TGFβ/ALK5-dependent cell motility.

We were able to show that treatment of Panc1 cells with MG132 increased the steady-state levels of SMAD7 protein resulting from an arrest in ubiquitin-mediated degradation of SMAD7. Next, we employed Panc1-RAC1B-KO cells to address the question of whether SMAD7 is targeted by RAC1B and found SMAD7 protein levels in these cells to be dramatically reduced under basal conditions (see Figure 3A). Moreover, the absence of RAC1B protein prevented the cells from upregulating SMAD7 protein in response to TGFβ1 treatment (see Figure 3B). The failure of Panc1-RAC1B-KO cells to respond to treatment with TGFβ1 with upregulation of SMAD7 protein could have contributed to the higher pSMAD3C:SMAD3 ratio and the prolonged SMAD3 activation observed earlier in these cells [28]. Since *SMAD7* itself is a TGFβ target gene that in most cell types is transcriptionally induced by this growth factor in an immediate–early fashion, we also monitored in a time-course analysis the response of *SMAD7* mRNA levels to TGFβ1 treatment. Surprisingly, Panc1-RAC1B-KO cells responded to challenge with TGFβ1 with a strong upregulation of SMAD7 mRNA which even exceeded that in the control cells, an effect that is likely due to higher ALK5 levels and a corresponding increase in activity of the pathway. The uncoupling of transcription and translation in the response of *SMAD7* to TGFβ stimulation in both RAC1B-KO cells suggests that control of SMAD7 expression and function by RAC1B is executed mainly at the level of protein stability.

Having shown that RAC1B induces SMAD7 protein expression, we sought to determine whether SMAD7 is able to mimic RAC1B’s inhibitory effects on ALK5 expression, TGFβ1/Smad-dependent transcriptional activity and target gene expression. To this end, ectopic overexpression of SMAD7 in Panc1-RAC1B-KO cells was able to reverse the high levels of ALK5 protein in these cells while knockdown of *SMAD7* in the corresponding LV control cells enhanced ALK5 protein expression (see Figure 4). In addition, transfected SMAD7 suppressed both TGFβ1-induced SMAD3-dependent transcriptional activity and *MMP2* and *SNAI2* mRNA expression in Panc1 cells (see Figure 4), while the reverse was true following RAC1B knockdown in Panc1 (see Figure 4) and MDA-MB-231 cells (see Appendix A).

To assess the functional consequences of SMAD7 induction by RAC1B for cancer-relevant cellular responses, we performed a series of real-time cell migration assays with a chemokinesis setup following KD/KO or ectopic overexpression of SMAD7. Knockdown of SMAD7, or RAC1B as control, resulted in a strong increase in TGFβ1-induced chemokinesis in both non-engineered Panc1 (see Figure 5) and MDA-MB-231 cells (see Appendix A) and in Panc1 or MDA-MB-231 cells with suppressed migratory potential due to stable or transient expression, respectively, of HA-RAC1B [22] (see Figure 5C and Figure 6C). Conversely, and as predicted from downregulation of ALK5 by MG132 treatment, this agent strongly inhibited TGF-β1 stimulated migratory activity of Panc1-RAC1B-KO cells (see Appendix A). Likewise, transfected SMAD7 quenched the TGF-β1-induced chemokinetic activity of (non-engineered) Panc1 or MDA-MB-231 cells, but not that of Colo357 cells (see Figure 6), and was able to completely or partially rescue Panc1 cells from the RAC1B knockdown/knockout-induced increase in cell migration (see Figure 7). Interestingly, in Colo357 cells that failed to downregulate ALK5 after MG132 treatment, or their migratory activity after SMAD7 transfection, SMAD7 is unable to interact with ALK5 [34], although an alternative mechanism of SMAD7 function was not supplied in this study.

Aberrant SMAD7 expression contributes to the invasion and metastasis of pancreatic cancer cells [35]. Of note, stable overexpression of SMAD7 in melanoma cells produced less bone metastases in a mouse model of bone metastasis and this effect was TGFβ-dependent [36]. Therefore, through its ability to increase SMAD7 protein expression RAC1B in an ALK5-dependent (or independent) manner may be able to decrease the cells’ sensitivity to TGFβ signaling and eventually prevent EMT and metastatic dissemination. In epithelial ovarian carcinoma cells SMAD7 has been proposed to maintain the epithelial phenotype and its inhibition results in mesenchymal transformation with increased expression of mesenchymal markers and decreased expression of epithelial markers such as E-cadherin [37]. Intriguingly, we have recently shown that RAC1B, too, promotes expression of E-cadherin [27], further supporting a functional interaction between RAC1B and SMAD7 in epithelial gene expression.

Prompted by strongly decreased mRNA levels of USP26 in Panc1-RAC1B-KD cells in conjunction with results from a previous study identifying this DUB as a regulator of SMAD7 [15], we pursued the idea that RAC1B might employ USP26 to increase SMAD7 protein abundance. Interestingly, we observed that *USP26* is transcriptionally induced by TGFβ1 in Panc1 cells in the same time frame as *SMAD7* and that this induction is lost in RAC1B-deficient cells. SiRNA-mediated knockdown of USP26 decreased SMAD7 levels, strongly suggesting that RAC1B utilizes USP26 for stabilization of SMAD7 protein and subsequent SMAD7-mediated degradation of ALK5. Also of note, despite their high sensitivity to TGFβ [22], Colo357 cells failed to upregulate USP26 mRNA after short-term exposure to this growth factor, providing a possible explanation for the inability of these cells to respond to MG132 treatment or SMAD7 transfection with a decrease in ALK5 protein levels or migratory activity, respectively. Our data are in line with those of Lui and coworkers who revealed for the first time the crucial role of USP26 in negative regulation of TGFβ signaling in MDA-MB-231 cells and cells from other tumor types. It will be interesting to see how modulating USP26 enzyme activity by genetic or pharmacological means will impact TGFβ1 and RAC1B-dependent cellular responses in PDAC and TNBC-derived cells. A schematic summary of the central roles of USP26 and SMAD7 in RAC1B control of TGFβ signaling is presented in Figure 9.

This study was designed to identify the series of events through which RAC1B targets ALK5 for inhibition rather than providing mechanistic details on the type of biochemical interactions among the various proteins of this pathway. Having shown that USP26 is induced by RAC1B and that its knockdown decreased SMAD7 protein abundance, it remains to be determined whether in pancreatic carcinoma cells USP26 indeed binds to and deubiquitinates SMAD7, as shown previously in breast cancer and glioma cells [15]. With the identification of *USP26* as a RAC1B target gene whose transcriptional activation is required for increasing SMAD7 protein it appears, however, less likely that RAC1B directly interacts with SMAD7. In addition, it needs to be elucidated whether SMAD7 complexes with activated ALK5 or, alternatively, recruits SMURF1 or SMURF2 to promote its ubiquitination and degradation.

While Panc1, IMIM-PC-1 and MDA-MB-231 cells are all representative of the (quasi-)mesenchymal subtype associated with poor differentiation (G3) and a c-EMT phenotype, Colo357 cells belong to the classical subtype associated with well-to-moderate differentiation (G1/G2) and a p-EMT phenotype [8]. Pancreatic tumor cells with c-EMT and p-EMT differ in their mode of cell migration (single cell vs. collective or single cell) and in the way they lose their epithelial phenotype. Although more cell lines of known subtype status need to be analyzed, i.e., MiaPaCa2, MDA-MB-157, MDA-MB-436, and MDA-MB-435s for the c-EMT arm and BxPC3, Capan2, MCF7, and MDA-MB-468 for the p-EMT arm [8], it is tempting to speculate that differences between the c-EMT and p-EMT programs extend to proteins other than E-cadherin and protein-protein interactions other than those involved in recycling and sequestration of E-cadherin into endosomes via Rab GTPases. Following this line of thoughts the ability of RAC1B to control ALK5 expression via USP26 and SMAD7 may operate only in mesenchymal-type cells with c-EMT but not in epithelial-like cells with p-EMT, i.e., Colo357 cells. A precedent in this respect is another Smad protein, SMAD3, which like RAC1B is expressed at much higher levels in well-differentiated as compared to poorly differentiated PDAC lines [29]. Moreover, much like SMAD7, SMAD3 is induced by RAC1B and in a non-C-terminally phosphorylated configuration also displays a potent antimigratory function in Panc1 [29] and MDA-MB-231 (H.U., unpublished observation), but not in Colo357 [38] cells.

The least differentiated tumors among PDAC and TNBC are characterized by worse prognosis and resistance to standard chemotherapy [2,5,7,9]. Even though, to date, the different strategies to subtype human PDAC and TNBC have mostly failed to improve therapies and patient outcome, these findings nevertheless suggest the existence of distinct treatment vulnerabilities in subgroups of patients. Therefore, a better understanding of the cell-autonomous mechanisms that distinguish mesenchymal from epithelial tumors may open opportunities for novel personalized therapies [3,4,5,39].

## 4. Material and Methods

### 4.1. Antibodies and Reagents

The following primary antibodies were used: anti-HSP90, #sc-7947 and #sc-13119, anti-TGF-β receptor I, V22, #sc-398, Santa Cruz Biotechnology (Heidelberg, Germany), anti-Rac1b, #09-271, Merck Millipore (Darmstadt, Germany), anti-β-actin, #A1978, Sigma (Deisenhofen, Germany), anti-GAPDH (14C10), #2118, Cell Signaling Technology (Frankfurt/Main, Germany), anti-SMAD7, #MAB2029, R&D Systems (Wiesbaden, Germany), anti-USP26, #MBS7044323, MyBioSource (San Diego, CA, USA). The HRP-linked anti-rabbit, #7074, and anti-mouse, #7076, secondary antibodies were from Cell Signaling Technology. Recombinant human TGFβ1, #300-023, was provided by ReliaTech (Wolfenbüttel, Germany). The proteasome inhibitor MG132 was purchased from Calbiochem/Merck and Protein A/G Sepharose from Santa Cruz Biotechnology (Heidelberg, Germany).

### 4.2. Cell Culture

Panc1 human PDAC cells were originally obtained from ATCC (Manassas, VA, USA) and cultivated in RPMI 1640 supplemented with fetal bovine serum, 1% Penicillin-Streptomycin-Glutamine (Life Technologies, Darmstadt, Germany) and 1% sodium pyruvate (Merck Millipore, Darmstadt, Germany). Colo357 and MDA-MB-231 cells were a kind gift from Dr. H. Kalthoff (Kiel, Germany). The PDAC-derived line IMIM-PC-1 was donated by Dr. Andre Menke (Gießen, Germany) and was originally generated by Dr. F.X. Real (Barcelona, Spain). IMIM-PC-1 cells are morphologically less differentiated and expresses low or undetectable levels of CK7 and MUC1 [40]. Colo357, Panc1 and MDA-MB-231 cells were maintained in RPMI and IMIM-PC-1 in DMEM, each supplemented with 10% fetal bovine serum, glutamine and penicillin/streptomycin. The generation of *RAC1* Exon 3b-deleted Panc1 cells by CRISPR/Cas9 technology and of Panc1 cell clones expressing HA-RAC1B in a stable fashion was described in detail earlier [22]. These cells were stably transduced using the pCGN vector, followed by selection of transduced cells with hygromycin and the isolation of individual cell clones by limited dilution. A pool of empty vector-transfected cells served as a control.

### 4.3. QPCR Analysis

Total RNA was extracted from Panc1 cells using PeqGold RNAPure from Peqlab (Erlangen, Germany) and purified according to the manufacturer’s instructions. For each sample, 2.5 μg RNA were subjected to reverse transcription for 1 h at 37 °C, using 200 U M-MLV Reverse Transcriptase and 2.5 μM random hexamers (Life Technologies) in a total volume of 20 μL. Relative mRNA expression of target genes was quantified by qPCR on an I-Cycler (BioRad, Munich, Germany) using Maxima SYBR Green Mastermix (Thermo Fisher Scientific, Waltham, MA, USA). Data were normalized to the expression of either TATA-box-binding protein (TBP) or GAPDH. For sequences of PCR primers see Appendix A.

### 4.4. Transient Transfection of siRNAs

On day 1 after seeding into plates (Nunclon^TM^ Delta Surface) from Nunc (Roskilde, Denmark) cells were transfected twice on two consecutive days with 50 nM each of control siRNA, RAC1B siRNA [22,23,25], SMAD7 siRNA (Silencer and SilencerSelect, Thermo Fisher Scientific) or USP26 siRNA {Dharmacon, Lafayette, CO, USA] and Lipofectamine 2000 (Life Technologies) according to the manufacturers’ instructions. In case of cotransfection of two different siRNAs, 25 nM each of the specific siRNA and 50 nM of control siRNA were used. Then, 24 h after the second transfection, cells were subjected to TGFβ1 stimulation followed by lysis in protein or RNA lysis buffer.

### 4.5. Cell Lysis and Immunoblotting

Confluent cells were washed once with ice-cold PBS and lysed with 1 x PhosphoSafe lysis buffer (Merck Millipore). Following sonication and clearing, the total protein concentration of the supernatants was determined with the BioRad DC Protein Assay. Samples were subjected to gel electrophoresis using BioRad mini-PROTEAN TGX any-kD precast gels and blotted to 0.45 μm PVDF membranes. Membranes were blocked with non-fat dry milk or BSA and incubated with primary antibodies at 4 °C overnight. HRP-linked secondary antibodies and Amersham ECL Prime Detection Reagent (GE Healthcare, Munich, Germany) were used for chemoluminescent detection of proteins on a BioRad ChemiDoc XRS imaging system.

### 4.6. Reporter Gene Assays

For the luciferase assays, cells were seeded in 96-well plates and were cotransfected on the next day serum free for 4 h using and Lipofectamine 2000 in combination with various siRNAs or expression vectors, the TGFβ/SMAD3-responsive reporter p(CAGA)_12_-luc (kindly provided by S. Dooley, Mannheim, Germany) and the Renilla luciferase encoding vector pRL-TK (Promega, Heidelberg, Germany). Then, 24 h later, cells were treated with 5 ng/mL TGFβ1 for another 24 h, followed by lysis and determination of luciferase activities with the Dual Luciferase Assay System (Promega). In all reporter gene assays, the data were derived from 6 wells processed in parallel and normalized with Renilla luciferase activity.

### 4.7. Real-time Cell Migration Assays

The xCELLigence^®^ DP system (ACEA Biosciences, distributed by OLS, Bremen, Germany) was employed for recording random cell migration of Colo357, Panc1, and MDA-MB-231 cells. CIM plates-16 were prepared according to the instruction manual and previous descriptions [22,23]. The underside of the upper chambers of the CIM plate-16 was coated with 30 μL of either collagen I (400 μg/mL) or a 1:1 (v/v) mixture of collagens I and IV. In all assays, RPMI with 1% fetal bovine serum was present in both the upper and lower chambers of each well of a CIM plate-16. The upper chamber of each well was loaded with 50,000–60,000 cells immediately after addition of 5 ng/mL TGFβ1 to the cell suspensions. Data acquisition was performed at intervals of 15 or 30 min and analyzed with RTCA software (ACEA).

### 4.8. Statistical Analysis

Statistical significance was calculated using the unpaired two-tailed Student’s *t* test, Mann–Whitney u or Wilcoxon test. Results were considered significant at *p* < 0.05 (*). Higher levels of significance were *p* < 0.01 (**) and *p* < 0.001 (***).

## 5. Conclusions

The data obtained in this study reveal that RAC1B effectively inhibits TGFβ1-dependent cell motility of mesenchymal subtype carcinoma cells by promoting protein expression of the inhibitory Smad, SMAD7, via intermittent transcriptional induction of the deubiquitinating enzyme, USP26. Therefore, therapeutically increasing RAC1B expression in poorly differentiated cancer cells from PDAC or TNBC may be a promising strategy to block the tumor-promoting functions of TGFβ1 and eventually enhance their redifferentiation to the less malignant epithelial phenotype.

## Figures and Tables

**Figure 1 cancers-12-01545-f001:**
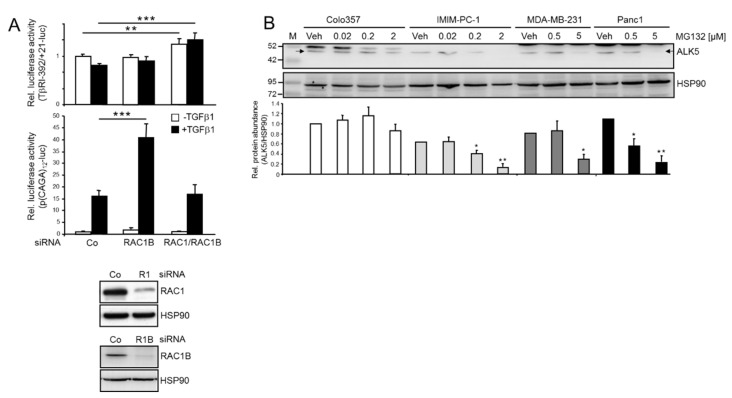
Negative regulation of ALK5 by RAC1B may involve regulation of ALK5 stability rather than downregulation of *TGFBR1* promoter activity. (**A**) Panc1 cells were cotransfected with 100 ng of either TβRI-392/+21-luc (upper graph) or p(CAGA)_12_-luc (lower graph), 25 ng pRL-TK-luc and 50 nM of either control (Co) siRNA, RAC1B siRNA or RAC1 siRNA (targeting both RAC1 and RAC1B). Then, 24 h later, cells were stimulated with TGFβ1 (5 ng/mL) for 24 h and subsequently processed for dual luciferase assay. Luciferase data (the mean ± SD of six parallel wells corrected for transfection efficiency with *Renilla* luciferase) are from a representative experiment performed three times. The asterisks indicate significance. The blots underneath the graphs show successful knockdown of RAC1 (R1) and RAC1B (R1B). (**B**) The PDAC cell lines Colo357, IMIM-PC-1 and Panc1, and the TNBC cell line MDA-MB-231 were treated for 24 h with the indicated concentrations of the proteasome inhibitor MG132, or vehicle (Veh) and subsequently processed for immunoblotting of ALK5 (specific band marked by arrows) and HSP90 as a loading control. The graph below the blot depicts data from densitometric readings of band intensities from underexposed autoradiographs and represents the mean ± SD of three experiments. Data are plotted relative to the respective Veh treated control cells set arbitrarily to 1.0. Significant differences (*p* < 0.05, two-tailed unpaired Student’s *t*-test) relative to the corresponding vehicle-treated control are indicated by an asterisk. M, molecular weight marker. * *p* < 0.05, ** *p* < 0.01, *** *p* < 0.001. Uncropped blots are shown in Appendix A.

**Figure 2 cancers-12-01545-f002:**
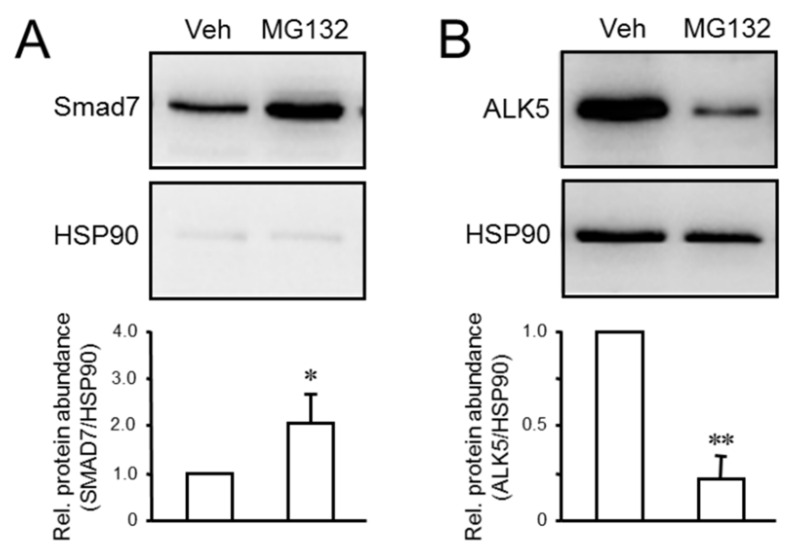
Effect of proteasome inhibition in Panc1-RAC1B-KO cells on SMAD7 and ALK5 protein levels. Panc1-RAC1B-KO cells were treated with the proteasome inhibitor MG132 (5 μM) or vehicle (Veh) for 24 h followed by immunoblot analysis of SMAD7 (**A**) and ALK5 (**B**). The graphs below the blots show data quantification from densitometric readings of band intensities from three independent experiments (the mean ± SD). The asterisks indicate significant differences. * *p* < 0.05, ** *p* < 0.01. Uncropped blots are shown in Appendix A.

**Figure 3 cancers-12-01545-f003:**
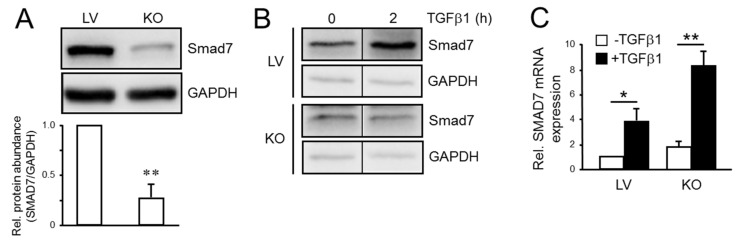
Effect of RAC1B knockout on SMAD7 expression in Panc1 cells. (**A**) Immunoblot analysis of SMAD7 expression in Panc1-RAC1B-KO (KO) and lentivector (LV) control cells under basal conditions. The blot was probed with an antibody to SMAD7 and to GAPDH as a loading control. The graph underneath the blot represents results from quantitative analysis based on densitometric readings of band intensities and represents the mean ± SD of three experiments. The asterisks indicate a significant difference (*p* < 0.01, Wilcoxon test). (**B**) As in (**A**), except that cells were treated, or not, for 2 h with TGFβ1. The thin vertical lines between bands indicate removal of irrelevant lanes. (**C**) Quantification of SMAD7 mRNA in Panc1-RAC1B-KO and LV cells by qPCR. Cells were treated for 1 h with TGFβ1 and subsequently processed for RNA isolation, reverse transcription and qPCR analyses of SMAD7 and β-actin. Data are the normalized mean ± SD of three wells processed in parallel and are representative of three experiments performed in total. The asterisks indicate significant differences (* *p* < 0.05, ** *p* < 0.01, unpaired two-tailed Student’s *t* test). Uncropped blots are shown in Appendix A.

**Figure 4 cancers-12-01545-f004:**
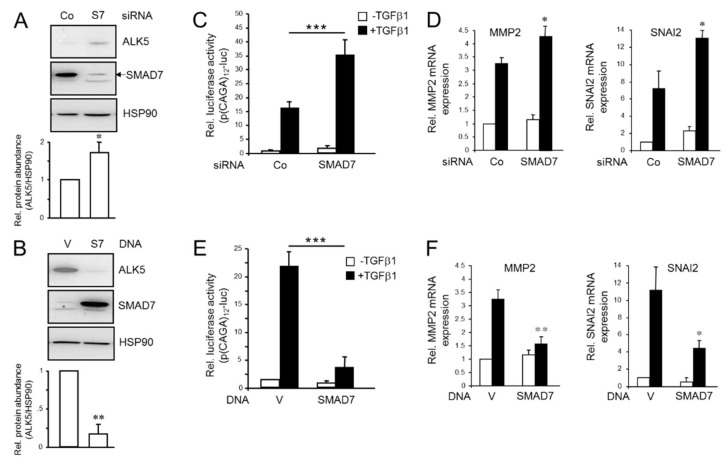
Effect of modulating SMAD7 expression on ALK5, TGFβ1-induced Smad-dependent transcriptional activity and target gene expression. (**A**) Panc1-LV cells were transfected twice on two consecutive days with irrelevant control (Co) siRNA or siRNA to SMAD7 (S7) and 48 h later processed for sequential immunoblotting of ALK5, SMAD7, and HSP90. (**B**) As in (**A**), except that Panc1-RAC1B-KO cells were used and transiently transfected with an expression vector for SMAD7, or empty vector (V) as control, The graphs underneath the blots show results from densitometry-based quantification of band intensities from underexposed blots derived from three independent experiments (the mean ± SD, *p* < 0.01, Wilcoxon test). (**C**) Panc1 cells were transiently transfected with Co siRNA or SMAD7 siRNA along with p(CAGA)_12_-luc and pRL-TK-luc, treated with TGFβ1 for 24 h and subsequently processed for dual luciferase assay. (**D**) As in (**C**), except that reporter genes were omitted and cells subjected to qPCR analysis of *MMP2* and *SNAI2*. (**E**) As in (**C**), except that the siRNAs were replaced by an expression vector for SMAD7, or empty vector (V) as control, respectively. (**F**) As in (**E**), except that reporter genes were omitted and cells processed for qPCR analysis of *MMP2* and *SNAI2*. Data in (**C**,**E**) are representative of four assays (the mean ± SD from 6 parallel wells). The asterisk indicates significance (*p* < 0.001, Mann–Whitney u test). Data in (**D**,**F**) are from three independent experiments (the mean ± SD, *n* = 3). The asterisk indicates significance (* *p* < 0.05, ** *p* < 0.01, *** *p* < 0.001, Wilcoxon test). Uncropped blots are shown in Appendix A.

**Figure 5 cancers-12-01545-f005:**
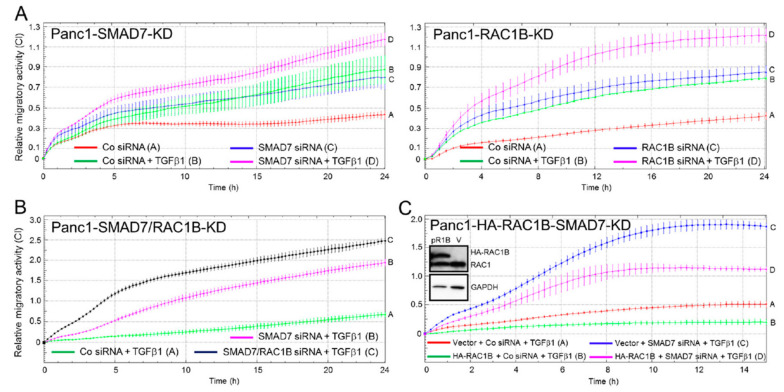
Effect of single or combined knockdown of SMAD7 or RAC1B on basal and TGFβ1-induced cell migration in Panc1 cells. (**A**) Panc1 cells were transfected twice on two consecutive days with 50 nM of irrelevant control (Co) siRNA and either SMAD7 siRNA (left-hand graph) or RAC1B siRNA (right-hand graph) as control. Then, 48 h later, the transfected cells were subjected to real-time cell migration assay (chemokinesis setup) in the absence or presence of TGFβ1. In both graphs, data show the mean ± SD of 3–4 wells per condition and are representative of three experiments with very similar results. Left-hand graph: Differences between Panc1 + Co siRNA + TGFβ1 (green curve, tracing B) and Panc1 + SMAD7 siRNA + TGFβ1 (magenta curve, tracing D) are significant the first time at 02:00 and all later time points. Right-hand graph: Differences between Panc1 + Co siRNA + TGFβ1 (green curve, tracing B) and Panc1 + RAC1B siRNA + TGFβ1 (magenta curve, tracing D) are significant at 04:00 and all later time points. (**B**) As in (**A**), except that Panc1 cells were transfected twice with 50 nM of Co siRNAs or 25 nM each of SMAD7 and Co siRNA, or 25 nM each of SMAD7 and RAC1B siRNA. Then, 48 h after the second round of transfection, cells were assayed for migratory activity on an xCELLigence platform in the presence of TGF-β1. (**C**) Panc1 cells stably expressing HA-RAC1B (pR1B) or empty pCGN vector (V) (verified by immunoblotting in inset) were transfected twice with either 50 nM of Co siRNA or SMAD7 siRNA. Only TGFβ-treated cells are shown. Differences between Panc1-vector + SMAD7 siRNA + TGFβ1 (blue curve, tracing C) and Panc1-HA-RAC1B + SMAD7 siRNA + TGFβ1 (magenta curve, tracing D) are significant at 02:00 and all later time points. Successful inhibition of SMAD7 and RAC1B expression was verified by immunoblotting (see Figure 4A and Figure 1A, respectively).

**Figure 6 cancers-12-01545-f006:**
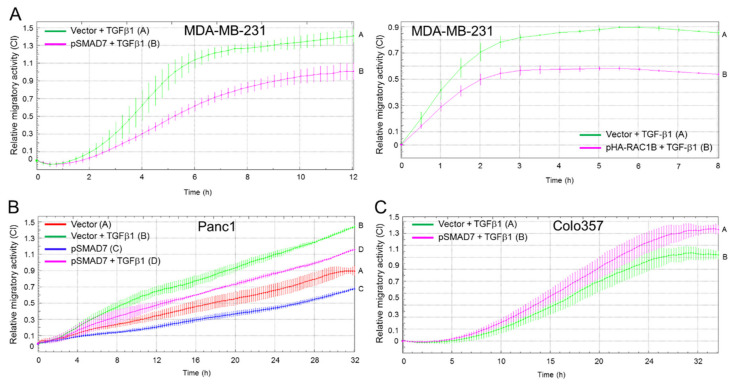
Effect of ectopic SMAD7 expression on TGFβ1-induced migration in Panc1, MDA-MB-231, and Colo357 cells. (**A**) MDA-MB-231 cells were transfected twice with either empty vector or a SMAD7 expression vector (pSMAD7, left-hand graph), or pHA-RAC1B (right-hand graph) as control, and 48 h later subjected to migration assay in the presence of TGFβ1. Differences between vector and SMAD7 transfected cells are first significant at 04:00 (left-hand graph) or 01:30 (right-hand graph) and all later time points. (**B**) Panc1 cells were transiently transfected with pSMAD7 or empty pcDNA3 vector and 48 h later subjected to real-time cell migration assay on the xCELLigence platform in the absence or presence of TGFβ1. Differences between vector + TGFβ1-treated cells (green curve, tracing B) and pSMAD7 + TGFβ1 transfected cells (magenta curve, tracing D) are first significant at 11:00 and all later time points. (**C**) As in (**A**, left-hand graph), except that Colo357 cells were used. The recorded migratory activities of SMAD7-transfected Colo357 cells were not significantly different from those of controls before the 20 h time point. In all panels, data are derived from one representative experiment and are the mean ± SD from 3–4 wells per condition. Note that in (**A**,**C**), only TGFβ1-treated cells are shown. Successful overexpression of SMAD7 in all three cell lines, and of HA-RAC1B in MDA-MB-231 cells, was verified by immunoblotting.

**Figure 7 cancers-12-01545-f007:**
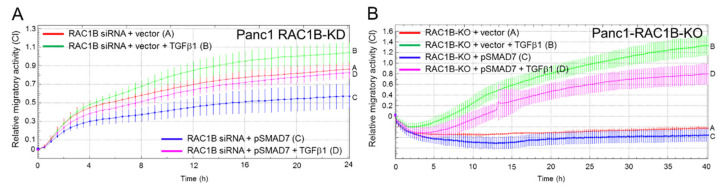
Effect of ectopic SMAD7 expression on TGFβ1-induced migration in Panc1-RAC1B-KD and KO cells. (**A**) Panc1-RAC1B-KD cells were generated by transfection with 50 nM of RAC1B siRNA on days 1 and 2, and on day 2, in addition, with either pSMAD7 or empty vector. Then, 48 h after the second round of transfection, cells were processed for migration assay. Data are the mean of 3–4 wells (the mean ± SD). Differences between Panc1 + RAC1B siRNA + vector + TGFβ1 (green curve, tracing B) and Panc1 + RAC1B siRNA + pSMAD7 + TGFβ1 (magenta curve, tracing D) are significant at 04:00 and all later time points. Successful inhibition of RAC1B and ectopic over expression of SMAD7 was verified by immunoblotting. (**B**) Panc1-RAC1B-KO cells were transiently transfected with either empty vector or pSMAD7 and 48 h later subjected to migration assay. Immediately before the start of the assay one half of the cells in (**A**,**B**) received 5 ng/mL TGFβ1. Data in each panel are from one representative experiment and are the mean ± SD from 3–4 wells per condition. Differences between Panc1-RAC1B-KO + vector + TGFβ1 (green curve, tracing B) and Panc1-RAC1B-KO + pSMAD7 + TGFβ1 (magenta curve, tracing D) are significant at 13:00 and all later time points. Successful inhibition of RAC1B and overexpression of SMAD7 was verified by immunoblotting.

**Figure 8 cancers-12-01545-f008:**
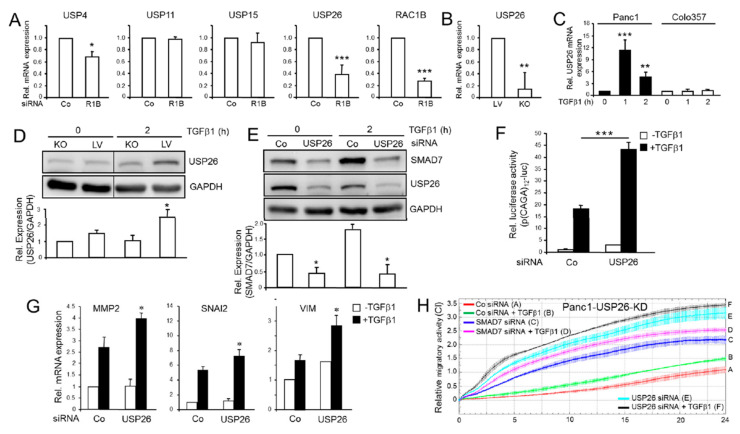
Effect of modulating USP26 expression on SMAD7, TGFβ1-induced transcriptional activity, target gene expression, and cell migration. (**A**) Panc1 cells were transiently transfected with control (Co) siRNA or RAC1B siRNA and 48 h later subjected to qPCR analysis for the indicated USPs. Data are the mean ± SD of at least three independent experiments (*n* = 3 for USP4, USP11 and USP15 and *n* = 7 for USP26 and RAC1B). (**B**) Cultures of Panc1-RAC1B-KO and LV cells were harvested at regular intervals and processed for qPCR analysis of USP26 (the mean ± SD, *n* = 5). (**C**) Panc1 or Colo357 cells were treated with TGFβ1 for the indicated times and USP26 mRNA levels determined by qPCR. Data are displayed as mean ± SD (*n* = 3). (**D**) Panc1-RAC1B-KO and LV cells were stimulated with TGFβ1 for 2 h and processed for immunoblot analysis of USP26. The graph below the blot shows the results from quantification of band intensities (the mean ± SD, *n* = 3). (**E**) Panc1 cells were transiently transfected with Co siRNA or USP26 siRNA, stimulated with TGFβ1 for 2 h and processed for immunoblotting of SMAD7, USP26, and GAPDH as loading control. The graph depicts results from data quantification (the mean ± SD) of three assays. (**F**) Panc1 cells were transiently transfected with or Co siRNA or USP26 siRNA, along with p(CAGA)_12_-luc and pRL-TK-luc, treated with TGFβ1 for 24 h and subsequently processed for dual luciferase assay. Data are shown as the mean ± SD of six parallel wells from a representative experiment performed three times. The asterisk indicates significance (*p* < 0.001, Mann–Whitney u test). (**G**) As in (**F**), except that reporter genes were omitted and cells subjected to qPCR analysis of *MMP2*, *SNAI2*, or *VIM*. Data were calculated from three experiments (the mean ± SD). (**H**) As in (**G**), except that Panc1 cells were subjected to real-time cell migration assay after transfection. Data are the mean ± SD of 3-4 wells and are representative of three experiments. Data are first significant at 01:00 and all later time points between USP26 siRNA + TGFβ1 and Co siRNA + TGFβ1 (black curve, tracing F vs. green curve, tracing B) and at 01:30 between SMAD7 siRNA + TGFβ1 vs. Co siRNA + TGFβ1 (magenta curve, tracing D vs. green curve, tracing B). The asterisks in (**A**–**E**) and (**G**) indicate significance (* *p* < 0.05, ** *p* < 0.01, *** *p* < 0.001, Wilcoxon test). The successful knockdown of *USP26* is shown in panel E.

**Figure 9 cancers-12-01545-f009:**
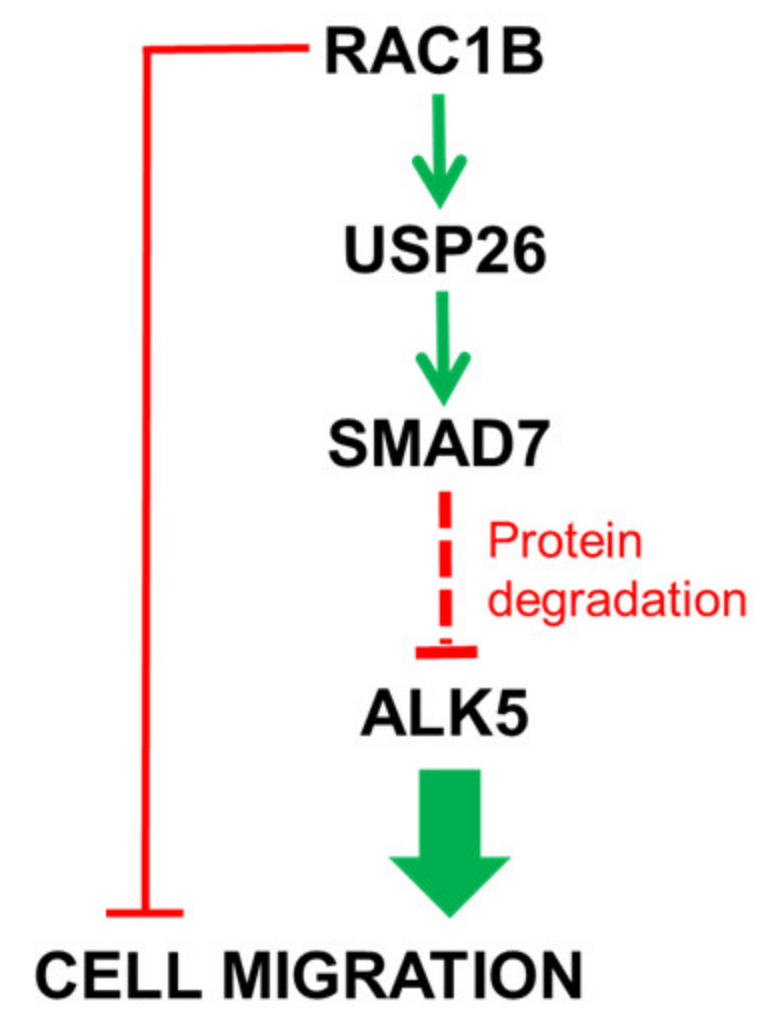
Schematic diagram illustrating the role of SMAD7 in negative regulation of ALK5 protein expression and cell migration by RAC1B. The suppressive effect of RAC1B on cell migration (red line on the left) involves induction of SMAD7 protein through intermittent induction of USP26 (green lines). As a consequence, the RAC1B-induced SMAD7 degrades ALK5 protein, eventually resulting in inhibition of TGFβ-induced cell migration. Stimulatory interactions are indicated by green arrows and inhibitory interactions by red lines.

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
