# Peer review of "RAC1B Induces SMAD7 via USP26 to Suppress TGFβ1-Dependent Cell Migration in Mesenchymal-Subtype Carcinoma Cells"

_cancers, 2020, doi:10.3390/cancers12061545_

Round 1

Reviewer 1 Report

The authors have addressed some of my concerns. However, considering the ongoing international problems it is understandable these concerns were not addressed. A few minor problems have been noted.

Line 91: Although reference 15 analyses these Dubs it is not the appropriate reference for this sentence.

The authors are recommended to include:

USP4 is regulated by AKT phosphorylation and directly deubiquitylates TGF-beta type I receptor. Zhang et al. 2012

USP11 augments TGFbeta signalling by deubiquitylating ALK5. Al-Salihi et al. 2012

USP15 stabilizes TGF-beta receptor I and promotes oncogenesis through the activation of TGF-beta signaling in glioblastoma. Eichhorn et al. 2012

USP26 regulates TGF-beta signaling by deubiquitinating and stabilizing SMAD7 Lui et al. 2017

The deubiquitinating enzyme UCH37 interacts with Smads and regulates TGF-beta signalling Wicks et al. 2005

Deubiquitinase activity profiling identifies UCHL1 as a candidate oncoprotein that promotes TGFbeta-induced breast cancer metastasis. Liu et al. 2019

The authors have also included the reference: TGF-β signaling pathway mediated by deubiquitinating enzymes. Kim et al. 2019.

It is also recommended they include some of the following.

Key role for ubiquitin protein modification in TGFβ signal transduction De Boeck et al. 2012.

Regulation of Ubiquitin Enzymes in the TGF-β Pathway Iyengar et al. 2017

Finally the use of LV is confusing and it would be clearer if these just said control or CTL. 

I have no need to see further revisions.

Author Response

We thank the reviewer for his understanding of the international situation which prevented us from satisfying all his comments, and appreciate his suggestion to include additional references.

1. A suggested, we have included the 6 recommended references (marked in yellow in the Reference list) and have accordingly renumbered all subsequent references in the text (also highlighted in yellow).

2. Finally the use of LV is confusing and it would be clearer if these just said control or CTL.

Response: We believe that the use of "LV" (for LentiViral-transduced negative control cells) is helpful as it prevents confusion with the siRNA control (designated "Co") and the empty vector control (designated "V") of the Smad7 expression vector. We would therefore prefer not to change it.

Reviewer 2 Report

This reviewer appreciates the extra effort of the authors to address the concerns raised. 

The authors have properly addressed most of the requirements and suggestions. 

Author Response

We thank the reviewer for his constructive comments which helped to seriously improve the quality of our manuscript.

Reviewer 3 Report

In the revised version, authors have addressed some of my concerns, and I find it is significantly improved overall. 

Author Response

We thank the reviewer for his constructive comments which helped to seriously improve the quality of our manuscript.

This manuscript is a resubmission of an earlier submission. The following is a list of the peer review reports and author responses from that submission.

Round 1

Reviewer 1 Report

In this manuscript, Ungefroren et al. seek to describe the role of RAC1B as a regulator of the TGFB receptor through the regulation of SMAD7. Using multiple cell lines the authors show that RAC1B can regulate SMAD7 stability. Furthermore, they demonstrate that genetic manipulation of SMAD7 alters the ability of RAC1B to regulate TGFB signaling. For the most part this is a well written article but is far too preliminary to warrant publication. Furthermore, there is significant overlap with both studies from the same group Witte et al (2017) and Otterbein et al (2019) decreasing the novelty of the current study. It is highly suggested that the authors discover the precise mechanism through which RAC1B regulates SMAD7 stability prior to future submissions. As indicated SMAD7 stability is regulated through multiple mechanisms including USP26, SMURF2, ITCH, and can also be sumoylated at multiple residues. Finding the exact function of RAC1B is this regard would be noteworthy. Other major and minor issues are as follows:

Major concerns:

  1. As discussed above the precise of role of RAC1B in this regard needs to be elucidated. Does RAC1B bind to SMAD7, SMURF2, or even the TGFB receptor. This has not been looked at.
  2. SMAD7/ SMURF2/ and TGFB receptor ubiquitnation needs to be analysed. Is this lysine 48 mediated ubiquitnation.
  3. Figure 1B. ALK5 qrt-pcr needs to be performed under the same conditions.
  4. Sentence 189. “ elevated ALK5 expression in Panc1-RAC1B-KO” is not shown. It is shown in figure 2D of Witte et al.
  5. Figure 3a needs to be performed in the presence and absence of MG132.
  6. Figure 3B there are only 2 bands how hard was to run these samples again. It questions the authenticity of this study.
  7. Figure 3C. This is a major concern as this figure shows the exact opposite results as to what was published in Witte et al. figure 8D.
  8. Figure 4. Again after performing the experiment 3 times. Is that really the nicest western blot you have?
  9. More importantly in figure 4. What really needs to be done is knockdown smad7 in these cells to show that ALK5 expression is retained. This would prove that SMAD7 is the target of RAC1B in the TGFB pathway.
  10. All of figure 5 has been published multiple times over the last 15 years by various groups. There is nothing significantly new about these results. Please demonstrate the relevance of RAC1B in relation to these results.
  11. Figure 6 and 7 are probably the nicest controlled experiments in this article but expression of proteins in these cells are not shown. You need to demonstrate that SMAD7 is knockdown and that RAC1B is knockdown or ectopically expressed.

Minor concerns:

  1. The second sentence in the abstract is very confusing it is recomended to rewrite this part.
  2. Sentence 122 required a reference.
  3. Sentence 133 paragraph requires justification.

Author Response

Major points

  1. As discussed above the precise of role of RAC1B in this regard needs to be elucidated. Does RAC1B bind to SMAD7, SMURF2, or even the TGFB receptor. This has not been looked at.

Response: We thank the reviewer for this suggestion. We fully agree that these are important questions that need to be answered. However, this study was designed to identify the series of events through which RAC1B targets ALK5 for inhibition rather than providing mechanistic details on the type of biochemical interactions among the various proteins of this pathway. We, therefore, have spared biochemical analyses, which will be analysed in more detail in a separate study. Given 4 factors (RAC1B, USP26, SMAD7, ALK5) this would require a large number of combinations to be tested in co-IP experiments which is beyond the scope of the current study. We should also mention that because of a shutdown (MRC-PPU) or limited capacity for shipping orders (Addgene) due to the COVID crisis, we were unable to obtain all the reagents, i.e. tagged versions of plasmids, required to carry out these experiments. This would have caused serious delays in performing these experiments within a reasonable period of time. Further, due to lower transfection efficiencies, co-IPs with plasmid DNAs in Panc1 or MDA-MB-231 cells are more difficult to perform than in easily transfectable cell lines such as HEK293T or NIH3T3 fibroblasts.

For these reasons and because this reviewer suggested to discover the precise mechanism through which RAC1B regulates SMAD7 stability, we attempted to clarify the mechanism, focussing on the role of USP26. We found that RAC1B induced USP26 in control but not in RAC1B knockout cells and that induction of this DUB was required for RAC1B-mediated upregulation of Smad7 protein. Moreover, siRNA-mediated knockdown of USP26 decreased basal and TGFβ1-induced SMAD7 protein levels and enhanced TGFβ1-induced reporter gene activity, gene expression and cell migration. All these new data on USP26 are presented in the new figure 8 in the revised version. 

  1. SMAD7/ SMURF2/ and TGFB receptor ubiquitination needs to be analysed. Is this lysine 48 mediated ubiquitination?

Response: Carrying out these experiments would require the use of HA-tagged ubiquitin and Lys-48 and Lys-63 ubiquitin mutants which are not available in our lab. For the same reasons specified above, we prefer to spare these data from our manuscript.

  1. Figure 1B. ALK5 qRT-PCR needs to be performed under the same conditions.

Response: As requested, we have assessed the expression of ALK5 by qRT-PCR. Our results confirm that except for MDA-MB-cells mRNA levels of ALK5 were not reduced by the MG132 treatment. The decrease in ALK5 mRNA levels in this cell line was, however, much lower that the decrease in protein levels.

  1. Sentence 189. “ elevated ALK5 expression in Panc1-RAC1B-KO” is not shown. It is shown in figure 2D of Witte et al.

Response: This is correct. We apologize for not having referred to our previous studies. The two references (#16 and #21) have now been included.

  1. Figure 3a needs to be performed in the presence and absence of MG132.

Response: We thank this reviewer for this suggestion. We like to emphasize that treatment of Panc1-RAC1B-KO cells with MG132 is shown in Figure 2A. Results indicate that Smad7 abundance was increased by 2-fold in KO cells.

  1. Figure 3B there are only 2 bands how hard was to run these samples again. It questions the authenticity of this study.

Response: Yes, this is correct. The two band represent the specific bands for SMAD7 and GAPDH (for verification of the correct sizes please see Suppl. data file with uncropped blots and molecular weight markers which I had supplied with the original submission). In this blot, irrelevant lanes have been removed as indicated by the thin vertical lines (now mentioned in the figure legend). We believe that this common approach significantly improves the graphical representation and should not question the authenticity of our results. As suggested, we have confirmed SMAD7 upregulation by TGFβ1 in parental Panc1 cells transfected with either scrambled control siRNA or USP26 siRNA (new Figure 8, panel E).

  1. Figure 3C. This is a major concern as this figure shows the exact opposite results as to what was published in Witte et al. figure 8D.

Response: It is true that in the Rac1b knockdown approach performed by Witte et al (Fig. 8D) did not demonstrate enhanced upregulation of Smad7 by TGFβ1 as presented here for the Rac1b knockout cells. We believe that, in order to obtain an increase in Smad7 mRNA abundance, expression of Rac1b needs to be downregulated in the cells to an extent that is only achieved in cells with complete knockout. In other words, the knockdown seen in Fig. 8D of the earlier publication was not efficient enough to cause this effect. We apologize for this discrepancy.

  1. Figure 4. Again after performing the experiment 3 times. Is that really the nicest western blot you have?

Response: We apologize for this shortcoming. We have replaced this blot with another one of the three blots used for quantification. We believe that this blot is more appealing and show even loading across all samples.

  1. More importantly in figure 4. What really needs to be done is knockdown smad7 in these cells to show that ALK5 expression is retained. This would prove that SMAD7 is the target of RAC1B in the TGFB pathway.

Response: As suggested, we have knocked down Smad7 in Panc1-LV cells by siRNA transfection (please note that Smad7 protein levels in -KO cells are too low for further inhibition) and consistently found that ALK5 expression is increased (new panel A in Figure 4). The original figure 4 has become panel 4B.

  1. All of figure 5 has been published multiple times over the last 15 years by various groups. There is nothing significantly new about these results. Please demonstrate the relevance of RAC1B in relation to these results.

Response: We agree with the reviewer that, in general, these results are not novel. However, to the best of our knowledge negative regulation of TGFβ1-induced p(CAGA)12-luc activity and MMP2 and SNAI2 expression by SMAD7 specifically in Panc1 cells has not been shown so far. We, therefore, believe that showing these results is crucial to demonstrate that SMAD7 can mimic the effects of RAC1B and would, thus, opt to leave this data in the manuscript. However, if requested by this reviewer, we are happy to move these data (panels C-F in the new figure 4) to the Supplementary data file.

  1. Figure 6 and 7 are probably the nicest controlled experiments in this article but expression of proteins in these cells are not shown. You need to demonstrate that SMAD7 is knockdown and that RAC1B is knockdown or ectopically expressed.

Response: We thank the reviewer for this suggestion. SMAD7 knockdown is shown now in Figure 4A, the RAC1B knockdown in Figure 1A (in response to comment #2 of Reviewer 2). Panc1 cells with stable overexpression of RAC1B have been characterized in detail previously (Ref. #16). However, we have included a RAC1B immunoblot in panel C of Figure 5 as an inset.

Minor points

  1. The second sentence in the abstract is very confusing it is recommended to rewrite this part.

Response: This sentence has been rephrased as suggested.

  1. Sentence 122 required a reference.

Response: The appropriate references referring to the cells’ sensitivity to TGFb (#15 and #16) have been added.

  1. Sentence 133 paragraph requires justification.

Response: Justification of this paragraph is now included in the revised version of the manuscript.

Reviewer 2 Report

In this paper, the authors study the effect of RAC1B on ALK5 inhibition through SMAD7 induction. The manuscript reports an interesting observation involving the effect of RAC1B-induced SMAD7 on inhibition of ALK5 protein stability without inhibition of transactivation of ALK5. Overall, the research is robust and well-doing. However, the manuscript as it currently stands raises some questions that need to be addressed.

Major comments:

  1. The authors showed that MG132 reduced the expression of ALK5 in IMIM-PC1, MDA-MB-231 and Panc1 cells, but not in Colo357 of Figure 1B. The results of the Western blot of Figure 1B represent two bands of ALK5. The authors explain that the upper bands of ALK5 in the supplementary figure are the non-specific bands. This reviewer thinks that the author should clarify whether the upper bands of ALK5 are a non-specific band using siRNA.
  2. The authors showed the luciferase activity of p(CAGA)-luc in Figure 1A. However, in the results, the authors did not show expression of RAC1B and RAC1 in siRNA-mediated down-regulation of RAC1B and RAC1 results. Therefore, the author should show the expression of RAC1B and RAC1 using Western blot or qPCR to confirm the results.
  3. The author should repeat the Western blot to clearly show the HSP90 in Figure 2A.
  4. The authors deleted RAC1B of the cells used in this study using CRISP/cas or lentivirus in Figures 2 and 3. Because the CRISP/cas or lentivirus systems are able to have off-target effects, to confirm the results in Figures 2 and 3, the authors should perform the same tests using shRNA to make stable cell lines. In addition, the author should show the expression of RAC1B using Western blot or qPCR in Figures 2 and 3
  5. To clarify the results in Figure 2, the author should perform Western blots at once to display the results.
  6. The authors should perform the Western blot repeatedly to clarify the signals of SMAD7 and HSP90 in Figure 4.
  7. Overall, in vitro studies were performed to show the results. This reviewer thinks that a simple experiment with a mouse model seems to be necessary for confirming the in vitro results.

Author Response

In this paper, the authors study the effect of RAC1B on ALK5 inhibition through SMAD7 induction. The manuscript reports an interesting observation involving the effect of RAC1B-induced SMAD7 on inhibition of ALK5 protein stability without inhibition of transactivation of ALK5. Overall, the research is robust and well-doing. However, the manuscript as it currently stands raises some questions that need to be addressed.

Major comments:

  1. The authors showed that MG132 reduced the expression of ALK5 in IMIM-PC1, MDA-MB-231 and Panc1 cells, but not in Colo357 of Figure 1B. The results of the Western blot of Figure 1B represent two bands of ALK5. The authors explain that the upper bands of ALK5 in the supplementary figure are the non-specific bands. This reviewer thinks that the author should clarify whether the upper bands of ALK5 are a non-specific band using siRNA.

Response: We thank the reviewer for raising this point. However, we believe that this issue has been conclusively addressed in our previous study cited as Ref. 16. In this study, we have performed experiments with ALK5 siRNA transfection. We have now included a file demonstrating specificity of the lower band. This figure is for convenience of the reviewer only and not for publication.

  1. The authors showed the luciferase activity of p(CAGA)-luc in Figure 1A. However, in the results, the authors did not show expression of RAC1B and RAC1 in siRNA-mediated down-regulation of RAC1B and RAC1 results. Therefore, the author should show the expression of RAC1B and RAC1 using Western blot or qPCR to confirm the results.

Response: As requested, we have performed immunoblots with a RAC1B and RAC1-specific antibodies. These blots have been inserted in Figure 1A of the revised version.

  1. The author should repeat the Western blot to clearly show the HSP90 in Figure 2A.

Response: As requested, we have replaced the HSP90 blot with another one with a stronger signal. However, the weak exposure of the HSP90 signal was used for quantifying band intensities since this requires underexposed images to make sure that signals are in the linear range.

  1. The authors deleted RAC1B of the cells used in this study using CRISP/Cas or lentivirus in Figures 2 and 3. Because the CRISP/cas or lentivirus systems are able to have off-target effects, to confirm the results in Figures 2 and 3, the authors should perform the same tests using shRNA to make stable cell lines. In addition, the author should show the expression of RAC1B using Western blot or qPCR in Figures 2 and 3.

Response: The Panc1-RAC1B-CRISPR/Cas cells have been extensively characterized for specific deletion of exon 3b in a previous publication. Please refer to Ref. 22 (Supplementary Figure S1) for details.

  1. To clarify the results in Figure 2, the author should perform Western blots at once to display the results.

Response: The signals for SMAD7 and HSP90 in panel A, and for ALK5 and HSP90 in panel B were derived from the same blots! For verification of the correct sizes please see Suppl. data file with uncropped blots and molecular weight markers which were included in the original submission.

  1. The authors should perform the Western blot repeatedly to clarify the signals of SMAD7 and HSP90 in Figure 4.

Response: This was also a request from Reviewer 1 (point #8). As requested, we have replaced this blot and renumbered this figure as panel B. In response to the request from Reviewer 1, we have also added immunoblot results on siRNA-mediated knockdown of SMAD7 on ALK5 protein expression in Panc1-LV cells as new panel A.

  1. Overall, in vitro studies were performed to show the results. This reviewer thinks that a simple experiment with a mouse model seems to be necessary for confirming the in vitro results.

Response: We agree with the reviewer that it would be nice to validate the results by in vivo investigations. However, this study was designed to clarify the molecular mechanisms through which RAC1B controls TGFβ signaling. Due to multiple pathways through which RAC1B exerts its tumor-suppressive function, selectively demonstrating the role of SMAD7 in a mouse model will be extremely challenging. Thus, the addition of in vivo investigations is currently beyond the scope of the present manuscript. In addition, due to the lock-down, our animal facility is currently closed.

Reviewer 3 Report

The study by Ungefroren et al. entitled “RAC1B induces SMAD7 to suppress the TGFb type I receptor ALK5 and TGFb1-dependent chemokinesis in mesenchymal-subtype carcinoma cells” is well organized and nicely written manuscript. The study further demonstrates the role of small GTPase RAC1B-induced SMAD7 driven negative regulation of ALK5 and TGFb1, which are key drives in tumor pathogenesis and EMT. I have the following concerns

Comparing Panc1-RAC1B-KO (KO) and lentivector (LV) control cells, authors must show immunoblot for successful KO.

For figure 5B, D authors must show immunoblots for successful KO and overexpression of SMAD7 and protein expression of MMP2 and SNAI2

Authors shall also show migration assay by other complementary experiments such as scratch-wound assay

Figures quality inserted in the manuscript is poor.

The title seems a little complex, shall be improved. 

The conclusions must be improved. 

Author Response

The study by Ungefroren et al. entitled “RAC1B induces SMAD7 to suppress the TGFb type I receptor ALK5 and TGFb1-dependent chemokinesis in mesenchymal-subtype carcinoma cells” is well organized and nicely written manuscript. The study further demonstrates the role of small GTPase RAC1B-induced SMAD7 driven negative regulation of ALK5 and TGFb1, which are key drives in tumor pathogenesis and EMT. I have the following concerns

  1. Comparing Panc1-RAC1B-KO (KO) and lentivector (LV) control cells, authors must show immunoblot for successful KO.

Response: The Panc1-RAC1B-CRISPR/Cas and LV control cells have been extensively characterized for specific deletion of exon 3b in a previous publication. Please refer to Ref. 22 for details.

  1. For figure 5B, D authors must show immunoblots for successful KO and overexpression of SMAD7 and protein expression of MMP2 and SNAI2.

Response: The immunoblots for successful knockdown and overexpression of SMAD7 are shown in Figure 4A and B, respectively. However, we believe that demonstrating changes in MMP2 and SNAI2 mRNA levels is sufficient to convince the reader that SMAD7 mimics the effects of RAC1B. For this reason, we would prefer not to include protein data for MMP2 and Slug.

  1. Authors shall also show migration assay by other complementary experiments such as scratch-wound assay.

Response: We believe that the xCELLigence data much more accurately reflect the actions of SMAD7 and RAC1B because this technique measures migration activity at the population level and data are highly quantitative. We think that performing scratch-wound assays would not provide additional information on the role of SMAD7 in RAC1B regulation of cell migration.

  1. Figures quality inserted in the manuscript is poor.

Response: We apologize for this shortcoming that must have occurred while the manuscript was converted to the PDF version. High quality images are now provided.

  1. The title seems a little complex, shall be improved.

Response: As requested, the title has been changed and now also accommodates the new data included in response to a request from Reviewer 1 (see new figure 8). 

  1. The conclusions must be improved. 

Response: This has been done.

Additional changes made:

  1. The former figures 4 and 5 have been combined to yield the new figure 4. The former panels 5A,B and 5C,D in Figure 5 have now become panels 4E,F and 4C,D, respectively, in the revised version. The numbering of the following figures has been changed accordingly.
  2. Figure 8 is new and contains the data on USP26 requested by Reviewer 1.
  3. Please note the change in the title.